# Impact of Oxygen Delivery on the Development of Acute Kidney Injury in Patients Undergoing Valve Heart Surgery

**DOI:** 10.3390/jcm11113046

**Published:** 2022-05-28

**Authors:** Elena Carrasco-Serrano, Pablo Jorge-Monjas, María Fé Muñoz-Moreno, Esther Gómez-Sánchez, Juan Manuel Priede-Vimbela, Miguel Bardají-Carrillo, Héctor Cubero-Gallego, Eduardo Tamayo, Christian Ortega-Loubon

**Affiliations:** 1BioCritic, Group for Biomedical Research in Critical Care Medicine, 47003 Valladolid, Spain; elenacarrascoserrano@gmail.com (E.C.-S.); pablojor@yahoo.es (P.J.-M.); jm.priede@gmail.com (J.M.P.-V.); mi321guel@hotmail.es (M.B.-C.); eduardo.tamayo@uva.es (E.T.); christlord26@gmail.com (C.O.-L.); 2Anesthesiology and Critical Care, Clinical University Hospital of Valladolid, 47003 Valladolid, Spain; 3Department of Surgery, University of Valladolid, 47003 Valladolid, Spain; 4Unit of Research, Clinical University Hospital of Valladolid, 47003 Valladolid, Spain; mfmunozm@saludcastillayleon.es; 5Interventional Cardiology Unit, Cardiology Department, Hospital del Mar, 08003 Barcelona, Spain; hektorkubero@hotmail.com; 6Centro de Investigación Biomédica en Red de Enfermedades Infecciosas (CIBERINFEC), Instituto de Salud Carlos III, 28029 Madrid, Spain

**Keywords:** Cardiac Surgery-Associated Acute Kidney Injury, minimum kidney oxygen delivery index, cardiopulmonary bypass

## Abstract

One of the strongest risk factors for death in individuals undergoing cardiac surgery is Cardiac Surgery Associated-Acute Kidney Injury (CSA-AKI). Although the minimum kidney oxygen delivery index (DO_2_i) during cardiopulmonary bypass (CPB) has been reported, the optimal threshold value has not yet been established. A prospective study was conducted from June 2012 to January 2016 to asses how DO_2_i influences the pathogenesis of CSA-AKI, as well as its most favorable cut-off value. DO_2_ levels were recorded at the beginning, middle, and end of the CPB. The association between DO_2_i and CSA-AKI was investigated using multivariable logistic regression analysis. The optimal cut-off of DO_2_i as a predictor of CSA-AKI was determined using Classification and Regression Tree (CART) analysis. A total of 782 consecutive patients were enrolled. Of these, 231 (29.5%) patients developed AKI. Optimal DO_2_i thresholds of 303 mL/min/m^2^ during the CPB and 295 mL/min/m^2^ at the end of the intervention were identified, which increased the odds of CSA-AKI almost two-fold (Odds Ratio (OR), 1.90; 95% CI, 1.12–3.24) during the surgery and maintained that risk (OR 1.94; 95% CI, 1.15–3.29) until the end. Low DO_2_i during cardiopulmonary bypass is a risk factor for CSA-AKI that cannot be ruled out. Continuous renal oxygen supply monitoring for adult patients could be a promising method for predicting AKI during CPB.

## 1. Introduction

Cardiac Surgery-Associated Acute Kidney Injury (CSA-AKI) is a severe complication and one of the strongest risk factors for mortality in patients after cardiac surgery, with an incidence varying from 8.9% to 42.5% according to the intricacy or complexity of the operation [1,2]. In its more severe form, it boosts the likelihood of death by 50–80% [3]. Even a mild rise in serum creatinine (SCr) after cardiac surgery can lead to a higher morbidity, as well as a longer term of stay in both hospital and intensive care unit (ICU) and a higher cost of care [4].

Non-modifiable predisposing variables for CSA-AKI include age, preoperative kidney insufficiency, female gender, low ejection fraction, diabetes, and emergency surgery. Modifiable factors include hemodilution, red blood cell transfusion, redo surgery, mean arterial pressure, postoperative hypotension, low cardiac output, cardiopulmonary bypass (CPB) period, and low oxygen delivery (DO_2_i) [5,6]. Low oxygen delivery is the amount of oxygen delivered to tissues throughout the body per minute, which is calculated from the cardiac output and arterial blood oxygen content. Hemoglobin content, oxygen saturation, pump flow, and arterial oxygen partial pressure all affect DO2 during CPB. When any of these components fall below critical levels, aerobic energy production can no longer keep up with oxygen requirements, triggering the anaerobic process to deliver energy to the cells and raise lactate levels. Innumerable efforts have been directed towards the control of modifiable risk factors [7]. Recently, the randomized goal-directed perfusion trial (GIFT), suggesting a novel goal directed perfusion (GDP) tactic, aimed to prevent a critical kidney oxygen delivery index lower than 280 mL/min/m^2^ diminished CSA-AKI in patients who had undergone moderately hypothermic CPB [8].

Despite all of this, not much is known about the effects of CPB on renal blood flow (RBF), the primary regulator of kidney oxygen supply (DO_2_). Renal vascular resistance increases from 15 to 23 percent during CPB as a result of the neuroendocrine response to CP. This includes increases in norepinephrine, vasopressin, and angiotensin II, causing blood flow to redistribute away from the kidneys [2]. Kidney hypoxia has been considered as a variable of paramount importance for the appearance of CSA-AKI [9,10]. The renal medulla is already on the verge of ischemia under normal circumstances [11]. This is related to a high medullary oxygen extraction rate and a poor perfusion rate, confirmed by the fact that the outer medulla has a tissue partial pressure of oxygen (pO_2_) of 10 to 20 mmHg, but the renal cortex has a pO_2_ around 50 mmHg [12,13]. Thus, the outer region of the kidney medulla is especially sensitive to low DO_2_ and is more vulnerable to harm, as commonly assessed through kidney tubular damage biomarkers. As a result, it is not unexpected that the kidneys could be one of the first organs to be damaged by a global decrease in DO_2_.

The minimum DO_2_i throughout CPB has been determined as an independent risk factor for dialysis [10] and the second state of Acute Kidney Injury Network (AKIN) [14]. The critical thresholds of DO_2_i have been identified to be 272 and 260 mL/min/m^2^, respectively. Nonetheless, none of these values are based on the novel CART analysis, which is a useful tool for classifying subjects into various risk categories and can partially overcome the risk of misclassification from the traditional Receiver Operating Characteristic (ROC) curve-based criteria to find the best cut-off value [15].

We hypothesize that normothermic CPB causes a diminished DO_2_i, leading to a kidney oxygen delivery/consumption imbalance. This study aims to determine the effects of DO_2_i on the appearance of AKI, in addition to determining the threshold value related to it.

## 2. Materials and Methods

### 2.1. Study Design

This investigation (identification number PI 19-1382) was authorized by both the Clinical Research Ethic Committee, which is a group of fifteen individuals, and the local Institutional Research Review Board.

We designed a prospective observational study to evaluate the effect of DO_2_i on CSA-AKI using the Kidney Disease: Improving Global Outcomes (KDIGO) criteria (i.e., an increment in SCr of ≥0.3 mg/dL within 48 h or an elevation in SCr to ≥1.5 times baseline, following 7 days after the intervention) [16]. Furthermore, we seek to determine the minimum DO_2_i threshold value related to CSA-AKI. We used the SCr levels recorded during the hospital entrance exam, as well as the peak SCr levels reached during the postoperative period of seven days.

This study took place in the Clinical University Hospital of Valladolid, a tertiary-level medical hospital with 800 beds, from June 2012 to January 2016. Approximately 450 open-heart interventions under CPB in adult patients are performed annually by the Cardiac Surgery Department, using two theaters on a daily basis. An intensive care unit (ICU) with 10 beds is dedicated exclusively for patients who have undergone cardiac surgery.

### 2.2. Study Patients

A total of 782 consecutive patients due to undergo elective heart surgery with cardiopulmonary bypass (CPB) were included in this study. Previous renal failure, off-pump surgery, urgent/emergency surgery, and patients with a kidney transplant were exclusion criteria [17].

### 2.3. Study Variables and Definitions

Population information included age, sex, body mass index (BMI), diabetes mellitus, hypertension, peripheral vascular disease, left ventricle ejection fraction (LVEF), logistic European System for Cardiac Operative Risk Evaluation (EuroSCORE), kidney depth, baseline bSo_2_, So_2_, preoperative hemoglobin, and preoperative SCr.

Perioperative variables collected included type of intervention; CPB duration; cross-clamp time; hematocrit (Hct); partial pressure of oxygen (pO_2_); oxygen delivery index (Do_2_i); and lactate at the beginning, during, and completion of CPB. 

Oxygen delivery was calculated using the formula [18]:DO_2_i = 10 × cardiac index (CI, L/min/m^2^) × oxygen content (mL/min/m^2^)

Oxygen content = (hemoglobin (g/dL) × 1.34 × oxygen saturation (%)) + (0.003 × pO_2_ (mmHg)), where hemoglobin, O_2_ saturation and pO_2_ were determined using blood gas analysis during CPB. Utilizing the same equation, the estimated DO_2_i for each patient was determined using the mean real time CI during CPB; the hemoglobin value; and the pO_2_ during the start, middle, and culmination of CPB.

Any degree of postoperative CSA-AKI as indicated by the KDIGO criteria was the primary endpoint.

### 2.4. Anesthesia and CPB Management

#### Intraoperative Care and Anesthesia

Intravenous midazolam and etomidate were used for the induction of anesthesia. Neuromuscular blockage was achieved using rocuronium. An oxygen/air mixture was used to ventilate the lungs to maintain normocapnia. A mixture of sevoflurane, midazolam and fentanyl was used to sustain anesthesia. To monitor arterial blood pressure and blood samples, a radial, brachial, or femoral artery catheter was placed. A trilumen internal jugular catheter was used to obtain central venous access for the assessment of central venous pressure, blood samples, and fluid/medication supply. An esophageal probe was used to detect the temperature. A Foley catheter was used to measure urine output.

Cardiac rhythm was continuously monitored. Systemic heparinization was achieved with 350 U/kg of heparin. To keep the active clotting time over 400 s, more heparin boluses were given. Pump output ranged between 45 and 55 mL/kg per minute. Intermittent cold blood cardioplegia and moderate systemic hypothermia (28–30 °C) were used. The mean arterial pressure was kept between 60 and 80 mmHg. Protamine was used to reverse the heparin effect at a 1.5:1 ratio after CPB completion and cannulae removal.

### 2.5. Statistical Analysis

According to a normal distribution, categorical data are represented as percentages, and continuous variables are reported as mean standard deviations (SDs) or medians (interquartile ranges (IQR)) as applicable. The Shapiro–Wilk and Kolmogorov–Smirnov tests were used to determine normality.

For categorical data, the relationship between CSA-AKI and other variables was identified using χ^2^, and for continuous data the Student’s *t*-test or the Mann–Whitney U test were utilized according to the normality criteria.

Data were used in a univariate logistic regression analysis to obtain 95% confidence intervals (CIs) for estimates. A stepwise approach was used to develop a multivariable logistic regression model. If the *p* value < 0.1, variables were included in the multivariable logistic regression model.

Regarding DO_2_, the optimal cut-off value for a higher risk for CSA-AKI was obtained using the Classification and Regression Tree (CART) Analysis, which is especially appropriate to the generation of clinical decision making. The capacity of this cut-off value to predict CSA-AKI development was further estimated using multivariate logistic regression analysis. The Hosmer–Lemeshow test was used to determine model calibration.

The odds ratio (OR) was calculated using a 95% confidence interval. Statistical significance was defined as *p* values <0.05. IBM SPSS Statistics for Windows version 24.0 was used to analyze the data (IBM Corp., Armonk, NY, USA). The R software was used to perform the CART analysis (Version 3.6.0. *R* Core Team, *R* foundation for Statistical computing, VIE, AU).

## 3. Results

A total of 231 (29.5%) patients developed CSA-AKI. In total, 48.7% of the individuals (n = 381) were female, and the median age was 70 (IQR, 63–77). Regarding the underlying conditions, the rates of hypertension, smoking, and EuroSCORE II were significantly higher in patients with CSA-AKI. They underwent longer CPB and cross-clamp periods, presented higher lactate levels, needed more red blood cell transfusions, and showed lower DO_2_i values throughout the procedure in comparison to those who do not develop CSA-AKI (*p* < 0.001). Table 1 shows the preoperative and intraoperative data.

CART analysis identified a DO_2_i < 303 mL/min/m^2^ during the CBP and <295 mL/min/m^2^ by the end of the intervention as the optimal cut-off values where there is already a risk for developing CSA-AKI (Figure 1).

The relationship between DO_2_i and CSA-AKI both during the surgery and at the end of the intervention is shown in Figure 2.

### Multivariable Model for DO_2_i and CSA-AKI

Univariate regression analysis revealed age (OR, 1.06; 95% CI, 1.03 to 1.07), CPB (OR, 1.02; 95% CI, 1.00 to 1.02), and aortic cross-clamp (OR, 1.01; 95% CI, 1.01 to 1.02) as risk factors for CSA-AKI. Continuous DO_2_ index was significantly associated with CSA-AKI (Table 2).

Table 3 reports the final multivariable model for DO_2_i and CSA-AKI. DO_2_i < 303 mL/min/m^2^ during the surgery (OR 1.90; 95% CI, 1.12 to 3.24) and DO_2_i < 295 mL/min/m^2^ (OR 1.94; 95% CI, 1.15 to 3.29) at the end of the intervention were significantly related to CSA-AKI. Other independent risk factors for CSA-AKI were age (OR 1.07; 95% CI, 1.03 to 1.11) and CPB (OR 1.01; 95% CI, 1.01 to 1.02) (Table 3). Mixing these data in a multivariable logistic model, an AUROC of 0.711 (95% CI: 0.670–0.751, *p* < 0.021) (Figure 2) was obtained.

## 4. Discussion

There were two major findings in this study. First, age, CBP time, lactate levels, and DO_2_i were significantly associated with CSA-AKI. Second, a decline in DO_2_i greater than 303 mL/min/m^2^ was correlated to postoperative CSA-AKI, as depicted in Table 2.

The incidence of CSA-AKI varies from 7% to 40%, depending on its numerous classifications and definitions [19]. In this study, 29.5% of the participants had CSA-AKI, which is similar to what is currently reported in the literature [1,2].

Multivariable logistic regression analysis identified that age, CPB time, lactate level, and DO_2_i were significantly associated with CSA-AKI. These data are supported by reported data, which show CSA-AKI etiopathogenesis to be highly complex [20]. Firstly, age is a well-known risk factor for the occurrence of CSA-AKI, which is used as a persistent and steady variable in several risk scores, such as the Age, Creatinine, and Ejection Fraction score; the EuroSCORE; and the Society of Thoracic Surgeons score [21]. Secondly, CPB triggers a complex series of reactions, resulting in the initiation of a systemic inflammatory response, the complement cascade activation, coagulopathy state and hemolysis, and the release of microemboli. The greater the extent of CPB is, the greater its consequences will be [22].

Kidney function is particularly dependent on O_2_ supply [23]. In 2005, Ranucci et al. were the first to identify a critical threshold for DO_2_i of 272 mL/min/m^2^ as a risk for acute kidney failure needing dialysis and postoperative peak SCr levels in 1048 patients who underwent coronary revascularization under CPB [10]. Later, De Somer et al. [14] showed that even a DO_2_i level < 262 mL/min/m^2^ is independently related to AKI. Most recently, in 2017, Magruder et al. [24] found that keeping DO_2_i > 300 mL/min/m^2^ during CPB lowered the occurrence of CSA-AKI. Likewise, Ranucci et al. determined, in the GIFT trial with 326 patients, that a DO_2_i < 280 mL/min/m^2^ increases the appearance of CSA-AKI [8]. Our findings are in line with the findings of the GIFT trial and provide robust support for the notion that keeping a DO_2_i above 303 mL/min/m^2^ is related to a decreased likelihood of developing CSA-AKI. The risk of CSA-AKI rises by 2% when the minimum DO_2_i is lower than this threshold value.

Typically, to achieve full-body perfusion, CPB perfusion flows of 2.0 to 2.4 L/min/m^2^ [10] or 2.5 to 3 L/min/m^2^ [24] at a mean blood pressure of 50 to 75 mmHg are appropriate. However, current attention was diverted from a fixed approach to a flow, based on DO_2_i, as a strategy to lower the occurrence of AKI after heart surgery. The exterior kidney medulla is already on the verge of hypoxia under normal circumstances. This kidney oxygen delivery/consumption mismatch, which is already present before CPB, is even more exacerbated during intervention. Lannemyr et al. [2] showed that despite keeping a systemic oxygen delivery during CPB, the kidney oxygen delivery/consumption balance was disrupted, and renal oxygenation was additionally impaired even after the CPB ended. This plays an essential role in the generation of tubular damage markers and AKI after heart surgery. The reduced renal oxygenation was induced by a drop in DO_2_ at a constant Vo_2_, an impaired arterial oxygen content due to hemodilution, and a re-distribution of blood flow away from the kidneys, despite an increase in systemic perfusion flow rate during CPB. This persistent state causes later ischemia-reperfusion kidney injury, leading to a lower kidney oxygenation and proximal tubular damage. Hence, predictors for AKI after heart surgery may not be precisely evaluated without taking into account the duration of low DO_2_ as a continuous variable.

Previous studies have shown the association between nadir DO_2_ and renal failure after surgery [10]. However, it must be highlighted that the data computed in those studies were collected intermittently at 10–20-min intervals. DO2 dynamically changes during CPB. Oshita et al. [25] found that both an extended and profound decline of DO_2_ under the cutoff value of 300 mL/min/m^2^ is preferable compared with established or fixed nadir DO_2_ parameters in predicting CSA-AKI. In addition, preventing hemodilution and excessive transfusion and improving CPB pump flow are of paramount importance to maintaining a DO_2_ over 300 mL/min per m^2^ [26]. Recently, Mukaida et al. [27] assessed both duration and intensity under the DO_2_ cut-off value and showed that an AUC lower than the DO_2_ of 300 mL/min/m^2^ and the cumulative time beneath that threshold value were more useful parameters for predicting CSA-AKI compared with nadir DO_2_. They found that when the aggregated time under 300 mL/min/m_2_ DO_2_ exceeded 15 min, the incidence of CSA-AKI increased considerably. This measurement included both momentary and continuous drops in DO_2_. However, while research on renal hypoxia has found that repeated transitory renal ischemia (for example, four periods of four minutes of ischemia for a total of 16 min of ischemia) does not significantly harm the kidneys, continuous ischemia for 20 min caused moderate kidney tubular injury due to ischemia reperfusion lesion [28]. Rasmussen and colleagues [29] showed that only an exposure to at least 30 min of DO_2_ < 272 mL/min/m^2^ was independently related to CSA-AKI.

Recently, Hendrix et al. [30] suggested that VO_2_ is variable and every patient has unique oxygen requirements, recommending the use of a personalized O_2_ supply target. Relying solely on one critical DO_2_ value for every individual is not reasonable. Without a patient’s VO_2_, nadir DO_2_ cannot be determined, which is dependent on the patient’s unique features and can be influenced by factors such as temperature and anesthesia technique, resulting in varying VO_2_ levels (ranging from 38 to 76 mL/min/m^2^) and, hence, unique DO_2_ demands for each patient. Thus, a perfusion goal focused solely on a generic fixed level for DO_2_ right over a universal critical value cannot always guarantee adequate DO_2_ for every patient. Furthermore, adjusting DO_2_ to the patient BSA allows for differences in physique, but does not consider distinctions in variables such as gender and age. As a result, oxygen demands during CPB are influenced by a variety of parameters, including the anesthetic method used and temperature regulation. Hence, VO_2_ should indeed be considered when addressing DO_2_.

Because VO_2_ cannot be maintained via aerobic energy generation at a crucial DO_2_ level, anaerobic metabolism is initiated to provide energy to the cells [14].

In a conscious human under normothermic conditions, the crucial Do_2_ nadir is approximately 300 mL/min/m^2^. In this study, we identified that under anesthesia and normothermia, when the Do_2_ is lower than 303 mL/min/m^2^ kidney function begins to deteriorate.

Several investigations have evaluated the association between mean arterial pressure during CPB and the development of AKI after heart surgery. Nevertheless, there are still conflicting data. While Ranucci et al. showed that hypotension during CPB may lead to a low DO_2_, raising the odds of CSA-AKI [31], Azau et al. found that raising mean arterial pressure does not decrease AKI after cardiac surgery [32]. Similarly, raising the mean arterial pressure did not improve renal function after surgery, according to Urzua et al. [33] and Sirvinskas et al. [34,35].

Patients with AKI had considerably lower levels of hemoglobin during CPB, which is understandable considering hemoglobin’s role in oxygen supply. A reduced DO_2_ may also be induced by the dilution of hemoglobin due to the prime solution used for the CPB circuit, usually a cell-free crystalloid solution [30]. The extent of hemodilution and a decreased systemic oxygen supply have been shown as independent predictors for the development of AKI after heart surgery [36]. Furthermore, transfusion may aggravate the risk of AKI [37]. This is because stored RBCs undergo a series of irreversible morphological and physiological alterations, including reduced oxygen affinity, shorter lifespans, and lower deformability. They are also more prone to damage [37,38]. As a consequence, RBC transfusions can precipitate renal injury due to reduced tissue DO_2_, oxidative stress activation, and the intensification of inflammatory responses [39,40].

Physiological indications could be employed to provide continuous bedside monitoring of the hemodynamics and functional processes involved in the development of CSA-AKI, rather than the current practice of sampling blood gas. Such kidney function measurements may permit the continuous real-time assessment of renal function just as the ventilation and hemodynamic are monitored, providing a functional approach for the application of kidney-protective strategies. Such monitoring could comprise kidney venous oximetry (which translates into the VO_2_) and the regional assessment of kidney oxygenation using near-infrared spectroscopy [20]. This shows that in order to deal with AKI, a protective tactic for AKI is necessary. Further advancement may be achieved with the development of continuous DO_2_i monitoring and the patient-specific prediction of individual DO_2_i thresholds.

### Study Limitations

First, we do not possess the Connect data management system (LivaNova), which can record the DO_2_i every 20 s to provide continuous real-time monitoring during CPB. We estimated the DO_2_i based on blood gas analysis. Second, DO_2_i was only calculated in three isolated periods of the CPB time. Third, during CPB we were not able to assess the time–dose effect of oxygen administration. Several researchers have defined the critical cut-off DO_2_i level by recording it every 10 to 20 min. Nevertheless, because the DO_2_i level is constantly and dynamically changing, it is possible that the reported DO_2_i levels do not accurately reflect a crucial condition. As a result, by analyzing nadir DO_2_i only, it is unclear whether appropriate DO_2_i was maintained during the CPB. Mukaida et al. identified that not only is maintaining levels of DO_2_i above specific threshold values important, but also the time below the nadir DO_2_i threshold level [27]. Thus, the understanding of the real consequences of DO_2_i throughout the entire CPB was not achieved.

## 5. Conclusions

The present data suggest that the appearance of renal failure after heart surgery is linked to DO_2_i during cardiopulmonary bypass. Low DO_2_i during cardiopulmonary bypass is a risk factor for CSA-AKI that cannot be ruled out. Continuous kidney oxygen supply monitoring may be a useful technique for preventing renal failure in adult patients undergoing cardiopulmonary bypass.

## Figures and Tables

**Figure 1 jcm-11-03046-f001:**
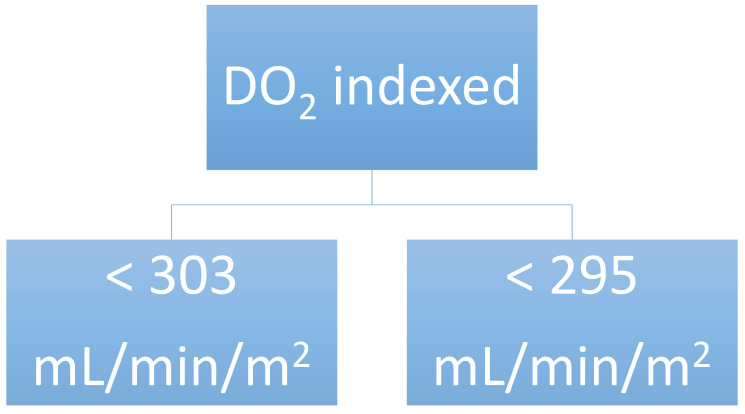
Decision-making analysis using the Classification and Regression Tree (CART) analysis, categorizing the kidney oxygen delivery index with an increased chance of developing acute renal injury after heart surgery.

**Figure 2 jcm-11-03046-f002:**
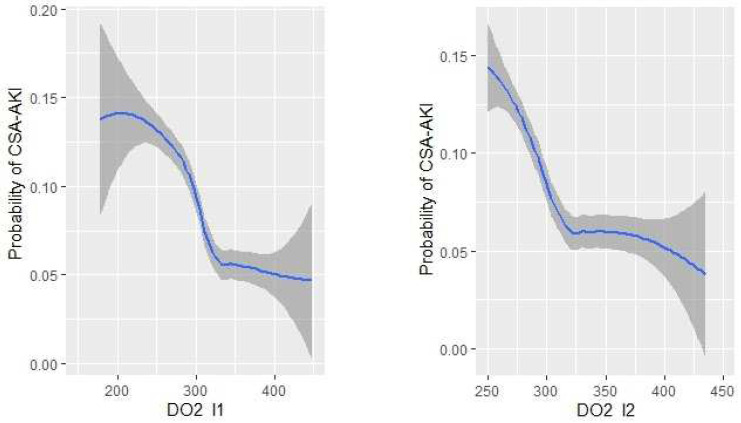
Relationship between DO_2_i and CSA-AKI both during the surgery and at the end of the intervention.

**Table 1 jcm-11-03046-t001:** Preoperative and intraoperative features based on CSA-AKI presence.

Characteristics	Total (n 782)	CSA-AKI (231)	CSA-AKI Free (551)	*p* Value
Preoperative data				
Population characteristics				
Age, y, median (IQR)	70 (63–77)	74 (67–78)	69 (62–76)	**<0.001**
Female	381 (48.7)	112 (48.5)	269 (48.8)	0.932
BMI, kg/m^2^, median (IQR)	27.0 (24.4–29.8)	27.1 (24.6–29.3)	27.0 (24.3–30.0)	0.923
EuroSCORE II, median (IQR)	1.7 (1.2–2.1)	1.8 (1.4–2.2)	1.6 (1.2–2.0)	**<0.001**
Smoker	163 (20.8)	52 (22.5)	111 (20.1)	**0.043**
Current Smoker	63 (8.1)	10 (4.3)	53 (9.6)	**0.043**
Hypertension	689 (88.1)	216 (93.5)	473 (85.8)	0.003
Diabetes mellitus	154 (19.7)	54 (23.4)	100 (18.1)	0.141
Dyslipidemia	574 (73.4)	176 (76.2)	398 (72.2)	0.253
COPD	43 (5.5)	12 (5.2)	31 (5.6)	0.533
Peripheral Vascular Disease	17 (2.2)	8 (3.5)	9 (1.6)	0.109
Prior Cardiac Surgery	49 (6.3)	19 (8.2)	30 (5.4)	0.339
Prior Stroke	36 (4.6)	17 (7.4)	19 (3.4)	0.106
AF	257 (33.0)	75 (32.3)	182 (33.0)	0.948
NYHA 3	230 (29.4)	73 (31.6)	157 (28.5)	0.371
NYHA 4	8 (1.0)	4 (1.7)	4 (0.7)	
Preoperative SCr, median (IQR)	0.8 (0.7–0.9)	1.21 (0.7–1.7)	0.95 (0.7–1.2)	**<0.001**
LVEF	62 (58–65)	62 (58–65)	62 (59–65)	0.740
Intraoperative data				
CBP time, min, median (IQR)	97 (80–123)	102 (83–131)	95 (80–117)	**<0.001**
Aortic Cross-Clamp time, min, median (IQR)	72 (57–93)	75 (60–99)	70 (56–91)	**<0.001**
Surgical procedure				
Aortic Surgery	374 (47.7)	105 (45.4)	269 (48.8)	0.007
Mitral Surgery	170 (21.7)	48 (20.8)	122 (22.1)	0.007
Mitral + Aortic Surgery	115 (14.7)	38 (16.5)	77 (14.0)	0.007
Mitral + Tricuspid Surgery	87 (11.1)	24 (10.4)	63 (11.4)	0.007
Mitral + Aortic + Tricuspid Surgery	24 (3.1)	11 (4.8)	13 (2.4)	0.007
Tricuspid Surgery	12 (1.5)	5 (2.2)	7 (1.3)	0.007
Red blood cell Transfusion	94 (12.0)	42 (18.2)	52 (9.5)	**<0.001**
Lactate, mg/dL	25 (19–32)	26 (21–33)	24 (19–32)	**<0.001**
DO_2_ indexed start CPB (mL min^−1^ m^−2^)	315.6 (283.0–350.2)	296.4 (330.6–271.3)	322.2 (290.3–357.3)	**<0.001**
DO_2_ indexed end CPB (mL min^−1^ m^−2^)	316.8 (284.8–354.9)	298.4 (272.1–330.8)	323.3 (293.0–351.7)	**<0.001**

AF, atrial fibrillation; CSA-AKI, Cardiac Surgery-Associated Acute Kidney Injury; BMI, body mass index; CABG, coronary artery bypass graft; COPD, chronic obstructive pulmonary disease; DO_2_, oxygen delivery; IQR, interquartile range; LVEF, left ventricle ejection fraction; MAP, mean arterial pressure; NYHA, New York Heart Association Classification; SCr, serum creatinine.

**Table 2 jcm-11-03046-t002:** Univariate logistic regression analysis for predisposing factors related to Cardiac Surgery-Associated Acute Renal Injury.

	Univariate Analysis	
Variables	OR (95% CI)	*p* Value
Age, y	1.06 (1.03–1.07)	<0.001
CBP time, min	1.01 (1.00–1.02)	<0.001
Aortic Cross-Clamp time, min	1.01 (1.01–1.02)	0.001
DO_2_ indexed during CBP (mL/min/m^2^)	0.99 (0.98–0.99)	<0.001
DO_2_ indexed end CBP (mL/min/m^2^)	0.99 (0.99–0.99)	0.002

CPB, cardiopulmonary bypass; DO_2_, oxygen delivery; OR, odds ratio.

**Table 3 jcm-11-03046-t003:** Multivariable logistic regression analysis for Cardiac Surgery-Associated Acute Kidney Injury using DO_2_ as a binary categorical variable. N = 782.

	Multivariable Analysis	
Variables	OR (95% CI)	*p* Value
Age, y	1.07 (1.03–1.11)	<0.001
CBP time, min	1.01 (1.01–1.02)	<0.001
DO_2_ indexed during CPB < 303 mL min^−1^ m^−2^	1.90 (1.12–3.24)	0.018
DO_2_ indexed end CPB < 295 mL min^−1^ m^−2^	1.94 (1.15–3.29)	0.014

CPB, cardiopulmonary bypass; DO_2_, oxygen delivery; OR, odds ratio.

## Data Availability

Data are available upon request to the corresponding author.

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
