# Peer review of "Impact of Oxygen Delivery on the Development of Acute Kidney Injury in Patients Undergoing Valve Heart Surgery"

_jcm, 2022, doi:10.3390/jcm11113046_

Round 1

Reviewer 1 Report

I have carefully read the manuscript of Carrasco-Serrano et al. and I found many interesting insights into understanding AKI in cardiac surgery. First of all, it is important to underline that the cutoff to be achieved during cardiopulmonary bypass is> 300 ml / min / m2 of DO2. Although DO2 in the entire study population is within the limits indicated by the literature, cases of AKi have occurred due to small variations in DO2. So it is necessary to underline the need to have different targets according to age, bypass duration and perfusion pressure. One threshold may not fit all. Again, it must be emphasized that the reduction of DO2 during cardiopulmonary bypass must be interpreted in terms of frequency, duration and severity. As for the reduction of pressure in anesthesia, it will be necessary to aim to evaluate the area under the DO2 threshold (AUT: 300 ml / min / m2) and the TWA weighted mean time of AUT. Only in this way will it be possible to take into consideration the real reduction value of DO2 and the impact on the outcome. About the preventive monitoring of kidney damage (doppler ultrasound monitoring of RBF, kidney venous
oximetry, and regional assessment of kidney oxygenation using near-infrared spectroscopy) I wouldn't be so confident. I see intraoperative Doppler evaluation as very complicated, if not impossible, as is NIRS in adults. I ask the authors to highlight these difficulties. For the rest, the manuscript has its own logic even if the limitations of a single-centered observational reduce the importance of the messages.

Author Response

Thank you for the suggestion. We have added the following sentences regarding this recommendation. We have deleted the doppler ultrasound monitoring of kidney blood flow as it is complicated to monitor intraoperatively.

Hendrix et al suggested that VO2 is variable, and every patient has an individual O2 needs, recommending for a personalized O2delivery goal. Using a single critical Do2 value for all patients is not sensible. Nadir Do2 cannot be evaluated without a patient’s Vo2, which as a result depends on the patient specific features and can be altered by factors such as temperature and anaesthesia technique, resulting in different Vo2 levels (ranging from 38 to 76 ml/min/m2) and hence particular Do2 demands for every patient.  Thus, a perfusion goal focused solely on a generic fixed value for Do2 just above a general critical Do2 level cannot always guarantee adequate Do2 for every patients.

References

Hendrix, R.H.J.; Ganushchak, Y.M.; Weerwind, P.W. Oxygen delivery, oxygen consumption and decreased kidney function after cardiopulmonary bypass. PLoS One 2019, 14, e0225541, doi:10.1371/journal.pone.0225541.

Reviewer 2 Report

The authors investigated the best cut off value for DO2i able to predict the occurrence of AKI.

The argument is not new, rather published in the late 90s by Ranucci and coworkers.

However, the study is well designed and Two major concerns:

1) authors refer to kidney-DO2i. Did the measured specifically the amount of oxygen delivered to the kidneys (it does not seem so from the manuscript!!!)? or did the calculated the overall DO2i according to Ranucci? In this latter case, please, remove kDO2i and rephrase all the manuscript accordingly.

2) in the univariate analysis, the authors introduce DO2i as a continuous variable. In the multivariate model, DO2i is included as a binary variable. Authors must declare which model has been followed.

Author Response

1) authors refer to kidney-DO2i. Did the measured specifically the amount of oxygen delivered to the kidneys (it does not seem so from the manuscript!!!)? or did the calculated the overall DO2i according to Ranucci? In this latter case, please, remove kDO2i and rephrase all the manuscript accordingly.

Response to reviewer

Thank you for the suggestion. We have already removed kDO21 from the manuscript and rephrased it accordingly.

2) in the univariate analysis, the authors introduce DO2i as a continuous variable. In the multivariate model, DO2i is included as a binary variable. Authors must declare which model has been followed.

Response to reviewer

Thank you for the question.  In the univariate analysis, DO2 resulted significant.  But in the multivariable analysis we wanted to specify which cut-off value of DO2 was significantly associated to CSA-AKI.  Therefore the cut-off value was used in the multivariable analysis.
